# Prevalence, Genetics and Evolutionary Properties of Eurasian Avian-like H1N1 Swine Influenza Viruses in Liaoning

**DOI:** 10.3390/v14030643

**Published:** 2022-03-20

**Authors:** Hailing Li, Haoyu Leng, Siqi Tang, Chaofan Su, Yina Xu, Yongtao Wang, Jiaming Lv, Shiwei Zhang, Yali Feng, Shaokang Song, Ying Zhang

**Affiliations:** Key Laboratory of Livestock Infectious Diseases, Ministry of Education, Key Laboratory of Zoonosis, College of Animal Science and Veterinary Medicine, Shenyang Agricultural University, 120 Dongling Rd., Shenyang 110866, China; 2018200144@stu.syau.edu.cn (H.L.); leng1071938690@gmail.com (H.L.); 2020220601@stu.syau.edu.cn (S.T.); 2020220609@stu.syau.edu.cn (C.S.); 2020220605@stu.syau.edu.cn (Y.X.); 2021220590@stu.syau.edu.cn (Y.W.); a18842422536@163.com (J.L.); z1148568393@163.com (S.Z.); 2020500031@syau.edu.cn (Y.F.); 18240397943@163.com (S.S.)

**Keywords:** swine influenza virus, Liaoning, phylogenetic, selection pressure, positive selection sites

## Abstract

Swine influenza virus (SIV) is an important zoonosis pathogen. The 2009 pandemic of H1N1 influenza A virus (2009/H1N1) highlighted the importance of the role of pigs as intermediate hosts. Liaoning province, located in northeastern China, has become one of the largest pig-farming areas since 2016. However, the epidemiology and evolutionary properties of SIVs in Liaoning are largely unknown. We performed systematic epidemiological and genetic dynamics surveillance of SIVs in Liaoning province during 2020. In total, 33,195 pig nasal swabs were collected, with an SIV detection rate of 2%. Our analysis revealed that multiple subtypes of SIVs are co-circulating in the pig population in Liaoning, including H1N1, H1N2 and H3N2 SIVs. Furthermore, 24 H1N1 SIVs were confirmed to belong to the EA H1N1 lineage and divided into two genotypes. The two genotypes were both triple reassortant, and the predominant one with polymerase, nucleoprotein (NP), and matrix protein (M) genes originating from 2009/H1N1; hemagglutinin (HA) and neuraminidase (NA) genes originating from EA H1N1; and the nonstructural protein (NS) gene originating from triple reassortant H1N2 (TR H1N2) was detected in Liaoning for the first time. According to our evolutionary analysis, the EA H1N1 virus in Liaoning will undergo further genome variation.

## 1. Introduction

Swine have been considered as an intermediate host of influenza virus, as they contain both avian-like receptors (α-2,3–linked sialic acids) and human-like receptors (α-2,6–linked sialic acids) in the up respiratory track epithelia cells, meaning that both avian and mammalian influenza viruses can replicate in swine [1,2]. Different subtypes of human and avian influenza virus have been isolated from pigs, such as H1, H2, H3, H4, H5, H7, and H9 subtypes [3,4,5,6], but only influenza virus of the H1 and H3 subtypes have established and started circulating in swine [7,8]. There are four main H1 subtypes of swine influenza virus (SIV) circulating in China: classical H1N1 (CS H1N1), Eurasian avian-like H1N1 (EA H1N1), triple reassortant H1N2 (TR H1N2), and 2009 pandemic H1N1 (2009/H1N1) [9]. CS H1N1 SIVs originated from the 1918 pandemic flu and were introduced into pigs in North America. EA H1N1 SIVs were transmitted from poultry to pigs in European countries in 1979 [10]. TR H1N2 was derived from CS H1N1, and triple reassortant H3N2 SIVs and emerged among North American pigs in 1998. During the 2009 human flu pandemic, 2009/H1N1 virus spilled over into the pig population and circulated. Consequently, reassortants between the EA H1N1 virus and 2009/H1N1 virus emerged in China and other countries [11]. Subsequently, the triple reassortant SIVs containing PB2, PB1, PA, NP, and M from 2009/H1N1; HA and NA from EA H1N1; and NS from TR H1N2 emerged in 2013 and have become predominant in most parts of China since 2016 [12].

China is one of the largest pork-consuming countries, with nearly 500 million pigs slaughtered annually. For environmental reasons, numerous large pig-farming companies in southern China have changed their expansion strategies by moving their pig farms to northeastern China since 2016. As a result, tens of large pig farms have been built in Liaoning province in recent years. As the pig population increases rapidly, the epidemiological situation of SIVs in Liaoning is becoming more and more complicated [13]. In this study, we performed extensive SIV surveillance in Liaoning province in 2020. We identified the genotypes and evolution characteristics of EA H1N1 SIVs in Liaoning. The EA H1N1 SIVs circling in Liaoning are under a high selection pressure. Our results indicated that the epidemiology of SIVs in Liaoning could become more complex in the future; thus, more attention should be paid to this situation in the northeast of China.

## 2. Materials and Methods

### 2.1. Sample Collection and Identification

During SIV active surveillance in 2020, 33,195 pig nasal swabs were collected from apparently healthy pigs from 30 different pig farms in 11 different cities of Liaoning. All the pig farms utilized all-in and all-out breeding methods. According to the biosecurity requirements, pig nasal swabs were collected from the finishing pig herds of each pig farm just before transferring them to slaughterhouses. In total, 66 finishing pig herds were sampled. The pig farms types, sizes, and sampling details can be found in Appendix A.

PCR based on influenza virus matrix (M) gene was conducted for the selection of positive samples [14]. Viral RNA of partial influenza-positive samples were extracted using a Viral RNA kit (Tiangen). cDNA was synthesized from vRNA by reverse transcription with influenza virus Uni12 primer [15]. All the primer information can be found in Appendix A. Each gene segment was amplified by PCR, with primers including each gene segment’s conserved promoter and noncoding region. PCR reactions were carried out under the following conditions: 94 °C for 2 min, followed by 30 cycles of 94 °C for 30 s, 57 °C for 30 s, 72 °C for 1 min, and a final extension at 72 °C for 2 min. The hemagglutinin (HA) and neuraminidase (NA) subtypes and 6 internal gene segments of the SIV isolates were determined by means of direct sequencing.

### 2.2. Hemagglutination Inhibition Assay

Pig sera were collected from herds of pigs of different ages, including suckling pigs, nursery pigs, gilts, sows, and boars, by pig farm resident veterinarians as part of routine surveillance, and the remaining sera were given to us for an SIV HI test. More pig farms and sampling details can be found in Appendix A. All the sera were treated with receptor destroy enzyme (Denka Seiken, Tokyo, Japan) according to the manufacturer’s instructions. The H1 subtype SIV standard antigen was kindly provided by the influenza group of the Harbin Veterinary Research Institute. A Hemagglutination Inhibition (HI) assay was performed as described elsewhere [16]. Briefly, 25 μL of eight HA units of the EA H1N1 antigen was mixed with 25 μL of two-fold serially diluted pig serum after incubating for 30 min. Then, 50 μL of 0.75% chicken red blood cells was added to each well, and the hemagglutination titer was recorded after 30 min [17,18,19]. 

### 2.3. Sequence Analyses 

The viral genome was sequenced by Sanger sequencing. The nucleotide sequences of our SIVs isolates were processed using the Lasergene sequence analysis software. The reference sequences for phylogenetic tree construction were retrieved from the 3 main influenza virus databases: the NCBI influenza virus database, the GISAID influenza database, and the Influenza Research database. The MEGA X [20] software was used for phylogenetic tree construction by applying the neighbor-joining method and 1000 bootstrap replicates [21]. The eight gene segments of Liaoning EA H1N1 SIVs were categorized into groups according to the 95% sequence identity cut-offs in the phylogenetic trees.

### 2.4. Selection Pressure and Positive Selection Sites Estimation

The selection pressure and positive selection sites of Liaoning EA H1N1 SIVs were estimated by calculating synonymous (dS) and non-synonymous (dN) substitution rates at every codon using the Datamonkey web server (http://www.datamonkey.org/), accessed 22 January 2022 [22]. In the current analysis, the Single-Likelihood Ancestor Counting (SLAC) and Mixed Effects Model of Evolution (MEME) methods were used to estimate the selection pressure [23,24]. The selection pressure was calculated by taking the average value of results estimated by two methods. The positive selection sites were estimated using MEME methods. For the selection pressure and positive selection sites analyses, the cut-off values for the *p*-value were set at 0.05.

## 3. Results

### 3.1. Epidemic of SIVs

During our SIV active surveillance program in 2020, nasal swabs of 33,195 pigs were collected from adult pigs in pig farms from 11 cities in Liaoning province as shown in Figure 1A. In total, 669 samples were confirmed to be influenza virus positive by RT-PCR test, and the detective rate reached 2.02% (95% CI 1.87–2.18%). For each sampling city, the detection rates of SIVs was from 1.27% to 2.60% as shown in Figure 1B. Shenyang, Chaoyang, Anshan, Dalian, Fuxin, and Tieling pig farms had higher detective rates above 2%, while other cities had lower detective rates that were still higher than 1% (Figure 1B).

We collected pig nasal swab samples from 66 finishing pig herds. SIV infections were found in 36 different herds, and one SIV-positive sample was chosen from one herd as the representative strain. The HA and NA genes of the 36 SIV-positive samples were sequenced and confirmed to be 24 H1N1 (Table 1), 11 H1N2, and 1 H3N2 subtype SIVs. The whole genomes of these H1N1 SIVs were further sequenced and classified into EA H1N1 lineages. EA H1N1 SIVs were detected in all sampling cities, whereas H1N2 SIVs were found in Shenyang and Jinzhou, and the H3N2 SIV was found in Shenyang. 

From same sampling cities but different pig farms, 2543 sera were collected from herd of pigs of diverse ages, including suckling pigs, nursery pigs, gilts, sows, and boars. The sera sampling instructions for each city are shown in Figure 2A. HI assay was performed to evaluate the seroprevalence of EA H1N1 SIVs in the Liaoning pig population. The pig sera with HI titer ≥80 were considered to be positive in order to avoid cross-antigen reactions. The pig sera HI titer ranges are shown in Figure 2A. 

As shown in Figure 2B, the seropositive rates of EA H1N1 virus showed great difference among the sampling cities. Anshan had the highest seropositive rate, which reached up to 99%, whereas Huludao had the lowest rate at only 3.33%. The seroprevalence of the pigs from each age group is shown in Figure 2B. The highest positive rate, which reached up to 54.89%, was found in the gilt group. The sow and boar groups seropositive rates were 44.38 and 36.84, respectively. The seroprevalence of the suckling and nursery pigs was >20%, but this was still lower than that of the other age groups. There were still pigs of unknown ages with a seroprevalence of 43.8%. Generally, the EA H1N1 SIV-seropositive rate of pigs in Liaoning was 36.8% (95% CI 34.94–38.72%).

### 3.2. Genetics of EA H1N1 SIVs

To understand the phylogenetic evolution of the EA H1N1 SIVs, the whole genomes of the 24 SIVs were analyzed. Their HA genes shared a 92.89–99.94% identity at the nucleotide level and formed three phylogenetic groups, whereas the NA genes shared a 92.62–100% identity and also formed three phylogenetic groups as shown in Figure 3. All the EA H1N1 SIV HA and NA genetic groups originated from the EA H1N1 SIVs.

The identity of the six internal genes was the polymerase basic protein 2 (PB2), polymerase basic protein 1 (PB1), polymerase acid protein (PA), nucleoprotein (NP), matrix protein (M), and nonstructural protein (NS) genes of the 24 viruses, at 94.17–100%, 94.90–99.96%, 93.90–100%, 93.30–100%, 91.85–100%, and 94.51–100%, respectively, at the nucleotide level. The NP and M genes formed two groups. The PB2, PB1, PA, and NS genes formed one group. The PB2, PB1, PA, and NP genes originated from the 2009/H1N1 lineage. Most SIV M genes originated from the 2009/H1N1 lineage except for one SIV (A/swine/Liaoning/DL1007/2020, DL1007), which originated from the EA H1N1 lineage. The NS group originated from the TR H1N2 lineage as shown in Figure 4. 

According to our genetic analysis results, the predominant genotype of EA H1N1 SIVs in Liaoning was a triple reassortant, which was recombinated with the EA H1N1, 2009/H1N1, and TR H1N2 lineages. Only one virus, DL 1007, belonged to a different genotype due to its M gene from the original EA H1N1 lineage. We named the two genotype groups G1 and G2, respectively.

### 3.3. Molecular Characters of EA H1N1 SIVs 

All the EA H1N1 viruses possessed a single basic amino acid in the HA cleavage site, PSIQSR/G, which was characteristic of low-pathogenic avian influenza virus. Mutations in the HA protein that contributed to the increased binding to human-type receptors, transmission, and replication, such as 190D (H3 numbering), 225E, and 226Q, were detected in the EA H1N1 SIVs [25,26,27,28,29]. A G to E mutation at the HA 158 site, which could alter the antigenicity of the virus [30], was also found in three viruses: PJ 521, PJ 626, and FS 1553 as shown in Table 2. 

The amino acids at positions 251, 271, 588, 431, 590, and 591 of the PB2 protein suggested that these viruses could adapt to mammalian hosts [13,31,32,33], but 627E and 701N indicated the relatively low pathogenicity of these viruses [34,35]. 

Other amino acid mutations, which could increase mammalian adaptation, pathogenicity, and anti-influenza drugs resistance, were also found in the NP, M2, and NS1 genes as shown in Table 2.

**Table 2 viruses-14-00643-t002:** Functional amino acid mutations found in Liaoning EA H1N1 SIVs.

Gene Segments	Amino Acid Mutation	Liaoning 2020 EA H1N1 SIVs
PB2		
A combination of 271A, 590S, and 591R plays a critical role in SIV replication and virulence in mammalian host [31]	PB2-T271A-A590S-A591R	PB2-271A-590S-591R
Increased viral replication and pathogenicity of EA H1N1 SIVs [32]	PB2-R251K	PB2-251R/K ^b^
Enhanced the virulence of EA H1N1 SIVs in mice [13]	PB2-T431M	PB2-431M
Enhanced 2009/H1N1 influenza virus virulence [33]	PB2-T588I	PB2-588I
Increased avian influenza virus adaptation in mammalian host [34,35]	PB2-E627K	PB2-627E
	PB2-D701N	PB2-701D
HA ^a^		
Change the antigenicity of EA H1N1 SIVs [30]	HA-E158G	HA-158E/G ^c^
Increase the receptor binding affinity to human-type a-2,6-linked sialic acid receptors [29]	HA-E190D	HA-190D
Increase the transmissibility of EA H1N1 SIVs in guinea pigs [25]	HA-G225E	HA-225E
NP		
A typical human signature marker, which enhances the pathogenicity of EA viruses in mice [36]	NP-Q357K	NP-357K
M2		
Resistance to adamantine derivatives [37]	M2-S31N	M2-31N
NS1		
The key amino acid in regulating the host IFN response and facilitating virus replication [38,39]	NS1-A42S	NS1-42S

^a^ The H3 numbering system was used. ^b^ Isolates with PB2-251R took up 62.5% (15/24), while those with PB2-251K took up 37.5% (9/24) of Liaoning 2020 EA H1N1 SIVs. ^c^ Isolates with HA-158E took up 12.5% (3/24), while those with HA-158G took up 87.5% (21/24) of Liaoning 2020 EA H1N1 SIVs.

### 3.4. Evolutionary Properties of EA H1N1 SIVs 

Given that the EA H1N1 SIV infection is common in the Liaoning pig population, we further analyzed its genomic evolution properties. There are only 33 Liaoning EA H1 SIVs sequences available online (including 29 EA H1N1 and 4 EA H1N2 SIVs) for now. The EA H1N1 SIV was first detected in Liaoning in 2006 and became prevalent gradually. Since 2014, reassortant EA H1N1 or H1N2 SIVs have been detected in Liaoning [13]. From 2014 to 2016, three different recombinated EA H1N1 SIVs have been identified, including two triple reassortants and one double reassortant, as shown in Figure 5. According to the emergence and reassortment of EA H1N1, we divided Liaoning EA H1 SIVs isolates into three time periods—from 2006 to 2013 (P1), from 2014 to 2016 (P2), and 2020 (P3)—to assess the adaptive molecular evolution characteristics.

The selection pressure of the HA and NA genes from P1 (11 HA and NA genes), P2 (22 HA genes and 18 NA genes except 4 N2 NA), and P3 (24 HA and NA genes) were analyzed. As shown in Table 3, the selection pressure of HA and NA during the P1 and P3 time periods was higher than in the P2 time period, indicating the more frequent occurrence of amino acid mutations in the HA and NA genes of SIVs in P1 and P3. For the isolates in the P3 G1 group, the selection pressure of internal genes ranged from 0.121 (NP) to 0.539 (NS1). 

There were three positive selected sites, 60, 219, and 361, in the HA gene across the P3 time period. HA 219 was located at the 220 loop; the mutation might affect the receptor binding property of the virus [40]. Positive selected sites were also observed in the PB2, PB1, PA, and NS1 genes of the SIVs in the P3 G1 group. There were five positive selected sites in PB1 at 14, 364, 607, 673, and 683, respectively. The amino acid substitution of PB1-V14A has been reported to influence the pathogenicity and transmissibility of avian influenza virus [41]. These isolates contained in PB1-14A took up 69.6% (16/23) of the G1 group, whereas PB1-14V took up 30.4% (7/23). PB2 was estimated to have two positive selected sites: PB2-391 locates at the cap-binding domain, and PB2-661 locates at the NP binding domain [42]. The PA and NS1 genes had two positive selected sites, respectively, at PA-436/532 and NS1-4/51, which, located at the domain, might affect the viral RNP activity [43] or the interaction with the host antiviral defense [44]. One positive selected site was located at NP-113, the biological function of which might need to be explored.

## 4. Discussion

In 2020, we performed extensive epidemiology surveillance of SIVs in Liaoning. Pig nasal swabs and sera were collected from 11 main pig-farming cities. Our results indicated that three subtypes of SIVs, namely H1N1, H1N2, and H3N2, are circulating in the pig population of Liaoning and that EA H1N1 SIV is the majority lineage. EA H1N1 SIVs in Liaoning evolved into two genotypes, G1 and G2, through recombination with the 2009/H1N1 and TR H1N2 virus. Serological surveillance indicated that the EA H1N1 virus exhibits different infection rates in different age groups of pigs. We found that these SIVs contain adaptation molecular characteristics to mammalian hosts and will acquire adaptive evolutionary mutation in the future.

The EA H1N1 SIVs began to circulate in China in 2001. A complex and extensive reassortment event occurred among EA H1N1, TR H1N2, and 2009/H1N1 SIV after the 2009 human influenza pandemic [45,46,47]. In our surveillance program, the EA H1N1 SIVs in Liaoning formed two genotypes, with most of the virus belonging to the G1 group: polymerase, NP, and M genes originating from 2009/H1N1; HA and NA genes originating from EA H1N1; and the NS gene originating from TR H1N2 except one virus in the G2 group, whose M originated from EA H1N1. The same recombinated EA H1N1 SIVs first emerged in southern China in 2013. The G1 virus became predominant in 2016 [12], whereas the G2 virus has not been found since 2018 in other parts of China. The G2 virus was predominant in Liaoning in 2015 [13], but the G1 virus was detected for the first time during our surveillance in Liaoning. It seems that the emergence and epidemic trend of SIV was comparably backward, which might be due to the lack of extensive surveillance in Liaoning province after 2016.

The SIV vaccine is rarely used in Liaoning pig farms, so serological studies will reflect the SIV infection rate. None of the sampled pig farms in our study applied the influenza vaccine either. There is no specific requirement concerning pig age for serological studies. Usually, we cannot obtain enough information about the infection rate among pigs of different ages. In our study, we collected sera from herds of pigs of various ages and found that the EA H1N1 SIV infection rate is related to the pig production stages. Suckling and nursery pigs had relatively lower EA H1N1 SIV infection rates, which might be due to their isolated breeding environment. Except for the unknown age of the herd, gilts had the highest SIV infection rate, which might be due to the breeding process of gilts. Generally, after the nursery stage, pigs that were healthy and met the breeding criteria were picked out from different herds and developed into gilt herds. The multiple sources of gilt herds might have contributed to their high EA H1N1 SIV seroprevalence. Sows and boars may have more SIV infection opportunities because they live longer than other herds.

Previous studies have shown that EA H1N1 SIVs are a great threat to human health: EA H1N1 can transmit efficiently among ferrets, and serological evidence has indicated that the EA H1N1 SIVs infection rates could reach 10.4% and 4.4% among swine workers and the general population [12,28,48]. In our study, multiple mammalian adaptation mutations already existed in the Liaoning EA H1N1 SIVs. Influenza virus is a type of virus with rapid mutation and evolution properties [49]. Beneficial mutations accumulation and genome segment recombination are the two main means of the influenza virus’s evolution. We performed an evolutionary pressure analysis of EA H1N1 SIVs. It was observed that the Liaoning EA H1N1 SIV genome is undergoing a relatively high natural selection pressure. The higher genome selection pressure indicates a faster evolutionary speed of the influenza virus [50]. According to this result, the triple reassortant EA H1N1 may acquire more mutations in the future and increase the public health risk.

## 5. Conclusions

Liaoning has a complex SIV ecosystem. Two genotypes of EA H1N1 SIVs are circulating in Liaoning, and more extensive mutations might be generated in EA H1N1 SIVs in Liaoning. The SIV epidemiology of Liaoning might be underestimated; thus, routine and extensive surveillance studies should be performed in Liaoning.

## Figures and Tables

**Figure 1 viruses-14-00643-f001:**
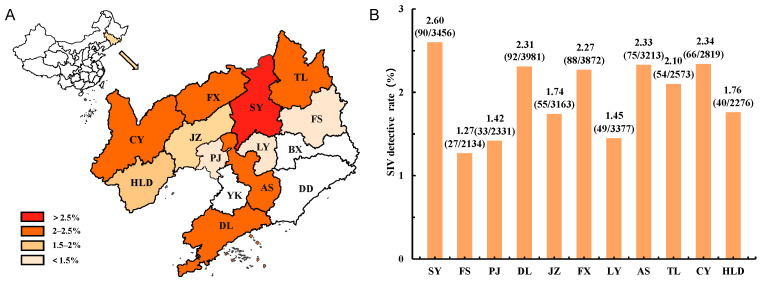
Detective of swine influenza virus (SIV) in Liaoning province. (**A**) Geographical distribution of sampling cities in Liaoning in 2020. The red color represents the isolation rate (%) of each city. (**B**) SIV detection rates in each city of Liaoning. SY, Shenyang; FS, Fushun; PJ, Panjin; DL, Dalian; JZ, Jinzhou; FX, Fuxin; LY, Liaoyang; AS, Anshan; TL, Tieling; CY, Chaoyang; HLD, Huludao.

**Figure 2 viruses-14-00643-f002:**
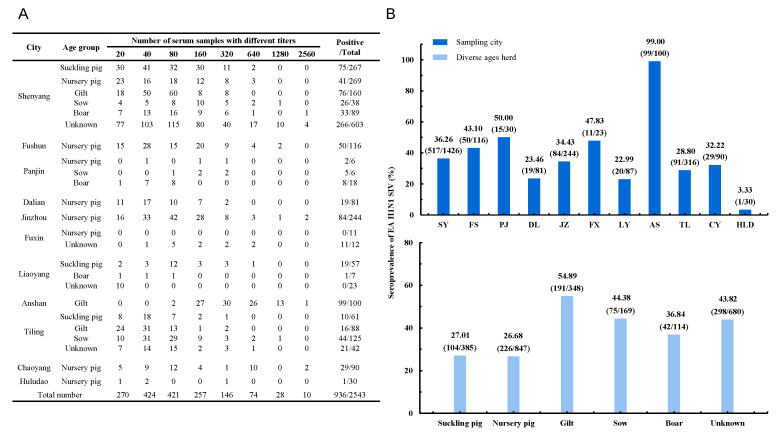
Seroprevalence of EA H1N1 SIVs in Liaoning province. (**A**) Sampling instructions for each city in Liaoning. (**B**). Seroprevalence of EA H1N1 SIVs in sampling cities and among herds of pigs of diverse ages in Liaoning. SY, Shenyang; FS, Fushun; PJ, Panjin; DL, Dalian; JZ, Jinzhou; FX, Fuxin; LY, Liaoyang; AS, Anshan; TL, Tieling; CY, Chaoyang; HLD, Huludao.

**Figure 3 viruses-14-00643-f003:**
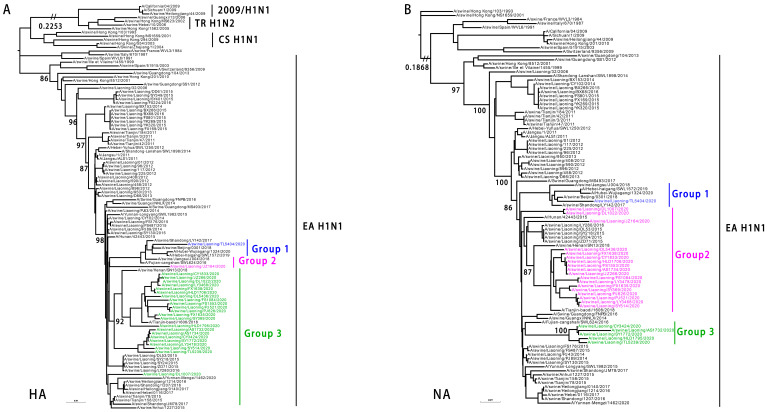
Genetic relationships among the HA and NA genes of EA H1N1 SIVs. (**A**). Phylogenetic tree of HA. The unrooted tree was based on nucleotides 33–1733. Sequences of viruses with names in black were downloaded from available databases; viruses with names in colors were sequenced in this study. (**B**) Phylogenetic tree of NA. The unrooted tree was based on nucleotides 21 to 1430. Scale bar indicates the number of nucleotide substitutions per site. 2009/H1N1, 2009 pandemic H1N1; EA H1N1, Eurasian avian-like H1N1; CS H1N1, classical swine H1N1; TR H1N2, triple reassortant H1N2.

**Figure 4 viruses-14-00643-f004:**
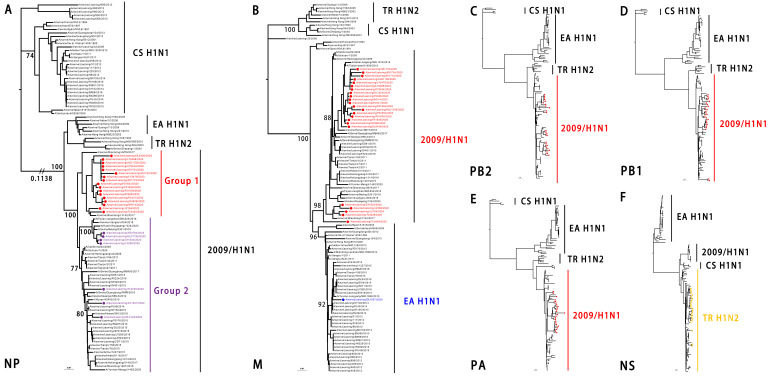
Genetic relationships among the internal genes of the EA H1N1 SIVs. (**A**). Phylogenetic tree of NP. The unrooted tree was based on nucleotide positions 46 to 1542; (**B**) 26 to 1007 for M; (**C**) 28 to 2307 for PB2; (**D**) 25 to 2298 for PB1; (**E**) 25 to 2175 for PA; and (**F**) 27 to 864 for NS. Scale bar indicates the number of nucleotide substitutions per site. 2009/H1N1, 2009 pandemic H1N1; EA H1N1, Eurasian avian-like H1N1; CS H1N1, classical swine H1N1; TR H1N2, triple reassortant H1N2.

**Figure 5 viruses-14-00643-f005:**
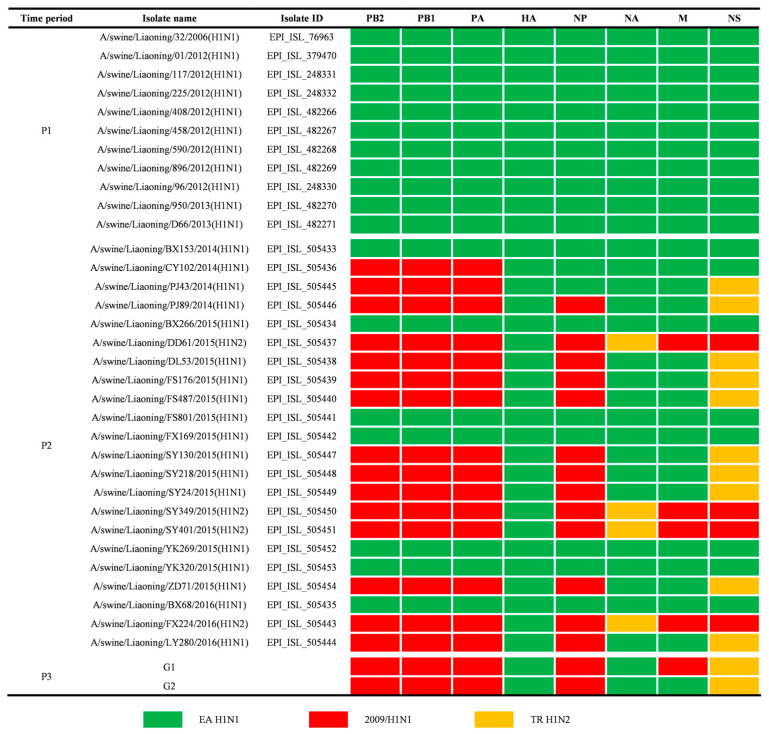
Genotypes of Liaoning EA H1N1 SIVs from 2006 to 2020. The eight gene segments are indicated at the top of each bar. The colors of the bars represent the three SIV lineages. Two genotypes of Liaoning 2020 SIVs were represented by G1 and G2.

**Table 1 viruses-14-00643-t001:** Information of Liaoning EA H1N1 SIVs.

Virus			Sample Information	
No.	Full Name	Abbreviation	Collected Date	Location
1	A/swine/Liaoning/JZ164/2020 (H1N1)	JZ164	19 January 2020	Jinzhou
2	A/swine/Liaoning/JZ266/2020 (H1N1)	JZ266	12 January 2020	Jinzhou
3	A/swine/Liaoning/SY369/2020 (H1N1)	SY369	14 February 2020	Shenyang
4	A/swine/Liaoning/SY514/2020 (H1N1)	SY514	11 September 2020	Shenyang
5	A/swine/Liaoning/PJ521/2020 (H1N1)	PJ521	25 March 2020	Panjin
6	A/swine/Liaoning/PJ626/2020 (H1N1)	PJ626	20 March 2020	Panjin
7	A/swine/Liaoning/DL1007/2020 (H1N1)	DL1007	21 April 2020	Dalian
8	A/swine/Liaoning/DL1022/2020 (H1N1)	DL1022	15 January 2020	Dalian
9	A/swine/Liaoning/FS1553/2020 (H1N1)	FS1553	17 May 2020	Fushun
10	A/swine/Liaoning/FS1084/2020 (H1N1)	FS1084	10 May 2020	Fushun
11	A/swine/Liaoning/FX1635/2020 (H1N1)	FX1635	11 June 2020	Fuxin
12	A/swine/Liaoning/FX1638/2020 (H1N1)	FX1638	18 June 2020	Fuxin
13	A/swine/Liaoning/HLD1706/2020 (H1N1)	HLD1706	27 July 2020	Huludao
14	A/swine/Liaoning/AS1732/2020 (H1N1)	AS1732	27 July 2020	Anshan
15	A/swine/Liaoning/AS1734/2020 (H1N1)	AS1734	9 August 2020	Anshan
16	A/swine/Liaoning/SY1772/2020 (H1N1)	SY1772	11 August 2020	Shenyang
17	A/swine/Liaoning/CY1833/2020 (H1N1)	CY1833	15 September 2020	Chaoyang
18	A/swine/Liaoning/HLD1795/2020 (H1N1)	HLD1795	15 September 2020	Huludao
19	A/swine/Liaoning/CY3424/2020 (H1N1)	CY3424	8 October 2020	Chaoyang
20	A/swine/Liaoning/LY3468/2020 (H1N1)	LY3468	3 August 2020	Liaoyang
21	A/swine/Liaoning/TL5239/2020 (H1N1)	TL5239	12 November 2020	Tieling
22	A/swine/Liaoning/LY3478/2020 (H1N1)	LY3478	12 November 2020	Liaoyang
23	A/swine/Liaoning/TL5404/2020 (H1N1)	TL5404	14 December 2020	Tieling
24	A/swine/Liaoning/DL5436/2020 (H1N1)	DL5436	14 December 2020	Dalian

**Table 3 viruses-14-00643-t003:** Analysis of the selection pressure of Liaoning EA H1N1 SIVs.

Gene Segment	Positive Selection Sites of 2020 EA H1N1 SIVs	Selection Pressure (dN/dS)		
		2020 ^a^	2014–2016	2006–2013
HA ^b^	60, 219, 361	0.193	0.160	0.215
NA ^c^	-	0.209	0.164	0.190
PB2	391, 661	0.125	-	-
PB1	14, 364, 607, 673, 683	0.135	-	-
PA	436, 532	0.163	-	-
NP	113	0.121	-	-
M1	-	0.243	-	-
M2	-	0.491	-	-
NS1	4, 51	0.539	-	-
NS2	-	0.253	-	-

^a^ Selection pressure and positive selection site assessments of the 2020 EA H1N1 SIVs internal genes (PB2, PB1, PA, NP, M1, M2, NS1, NS2), not including DL1007. ^b^ Selection pressure of 11 HA genes from P1, 22 HA genes from P2, and 24 HA genes from P3. The H3 numbering system was used. ^c^ Selection pressure of 11 NA genes from P1, 18 NA genes (except 4 N2 NA) from P2, and 24 NA genes from P3.

## Data Availability

The eight gene segments of the isolates were sequenced and have been deposited in GenBank under the following accession numbers: OL310825.1–OL310845.1, OL310856.1–OL310858.1, OL310925.1–OL310947.1, OL310953.1, OL310965.1–OL310988.1, OL310991.1–OL311014.1, OL311114.1–OL311137.1, OL311360.1–OL311383.1, OL311392.1–OL311415.1, OL314755.1–OL314778.1.

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
