# Peer review of "Prevalence, Genetics and Evolutionary Properties of Eurasian Avian-like H1N1 Swine Influenza Viruses in Liaoning"

_viruses, 2022, doi:10.3390/v14030643_

Round 1

Reviewer 1 Report

The manuscript by Li, et al. describes molecular and genetic characterizations of swine influenza A viruses (SIV) in circulation in herds in 2020 in Liaoning, province in the Northeastern of China. The team also provides results of serological survey conducted at the same period. The authors focus on the genetic variability of Eurasian avian-like H1N1 SIV that infected pigs and exhibit selection pressure on SIV. Please find below some remarks/suggestions.

Comments:

Line 59: English language, “should be paid” isn’t it? Throughout the manuscript, please check singular/plural agreements.

  • 2 - Sample collection:

Authors should explain representativeness of sampling to define prevalence and apply a confidence interval to positive results. How was calculated the sampling for this virological active surveillance? The reader also need more explanation about the serological sampling here. Pig sera from different farms is an information read in the results part. What are the types of farms, their size and influenza vaccination programs? What proportion of the total pig population in Liaoning province represent the sampling?

Line 65: why the term “partial” in ‘Viral RNA of partial influenza positive samples’? Indicate the proportion of full genome sequenced viruses.

Line 67: the authors explain they used only Uni12 primer, without Uni13?

Line 70: I don’t understand “by means” in the sentence.

Lines 74-75: please, name the antigen used in the HI test, confirm the concentration of chicken red blood cells in the protocol (0.75%), because it could be 0.50%. As reviewer, it would be like to have access to the detailed results (HI titres).

Line 86: is there a mistake with the word MAGA X? perhaps MEGA X?

Line 95: “were performed” instead of “was performed”?

  • 3 - Results:

Figure 1A: the legend of the figure A is incomplete (unit…)

Line 124: can the authors explain the threshold of HI titer of 80 for considering positive serum? As HI titers are available upon request, can the authors give more information, i.e. the range of HI titers, the mean…

Line 134: a comma missing

L135: plural sentence

L140-143: to be revised, not really agree. It’s hard to make a relationship between seroprevalence and positive detection without understanding the sampling of the active surveillance. Moreover, such explanation should be in the discussion part.

Figure 3 and 4: the resolution must be improved because we cannot read the name of the viruses when enlarging the tree. The legend of the figure 4 must be revised. A sentence is written twice.

L158-165: please, revise the English language (singular/plural…)

L176-177: “due to its M gene”, without ‘s’. Why the authors did not appropriate the genotypes nomenclature by Sun [12]? G1 and G2 are not new in China, and correspond to G4 and G5 respectively. G1 is also the same genotype called G14 by Xu [13], isn’t it?

L182: add ‘in’ in the sentence “… had been detected in the EA H1N1 SIVs.”

L179-180: specify that HP/LP is specific to the avian viruses adding, “which was characteristic of low pathogenic avian influenza virus”.

L184: add the reference to table 2 at the end of the sentence.

Table 2: please modify Ha-E158G by HA-E158G

 L211: move the comma after genes and not after NA

L215: remove “and positive” in this sentence?

Discussion

L241: modulate the idea with the conditional tense

L274-275: are there data of dN/dS obtained from other influenza viruses (other subtypes/lineages) to compare the values and discuss this level of selection pressure?

Conclusion

L282: twice “under”

Author Response

Response to Reviewer 1 Comments

Comments:

Point 1: Line 59: English language, “should be paid” isn’t it? Throughout the manuscript, please check singular/plural agreements.

Response 1: The “should be pay” had been changed to “should be paid”

2 - Sample collection:

Point 2: Authors should explain representativeness of sampling to define prevalence and apply a confidence interval to positive results. How was calculated the sampling for this virological active surveillance? The reader also need more explanation about the serological sampling here. Pig sera from different farms is an information read in the results part. What are the types of farms, their size and influenza vaccination programs? What proportion of the total pig population in Liaoning province represent the sampling?

Response 2:

The confidence intervals results had been added.

According to the biosecurity requirements, pig nasal swabs were collected from finishing pig herds of each pig farms just before transferring them from pig farms to slaughterhouses. Because all the pig farms performed all-in and all-out method, all the sampling pigs were from a same herd in a same building. The circulating SIVs in the all-in and all-out herd should be quite similar. We chose 1 isolate from each herd as the representative strain. The types of farms, their size and more explanation about the sampling had been added in supplement Table S2 and S3.

None of the pig farms applied the influenza vaccine and no vaccination data was available. According to the National Bureau of Statistics of China, in 2020, the number of finishing pigs in Liaoning province was 22.4 million (http://www.stats.gov.cn/tjsj/ndsj/2020/indexch.htm). Our sampling pigs took up 0.15% of the total finishing pigs in Liaoning province.

Point 3: Line 65: why the term “partial” in ‘Viral RNA of partial influenza positive samples’? Indicate the proportion of full genome sequenced viruses.

Response 3: 669 samples had been confirmed to be influenza positive by PCR and we sequenced 36 representative isolate, which were partial of the positive samples.

Point 4: Line 67: the authors explain they used only Uni12 primer, without Uni13?

Response 4: Uni12 primer only could effectively help the single-strand cDNA synthesis and completed the influenza virus RNA reverse transcription progress.

Point 5: Line 70: I don’t understand “by means” in the sentence.

Response 5: The “by means” had been changed to “by method”

Point 6: Lines 74-75: please, name the antigen used in the HI test, confirm the concentration of chicken red blood cells in the protocol (0.75%), because it could be 0.50%. As reviewer, it would be like to have access to the detailed results (HI titres).

Response 6: The antigen name was “EA H1N1 swine influenza virus HI test standard antigen”. The antigen was provided by Influenza group of Harbin Veterinary Research Institute.

We used 0.75% red blood cells in both hemagglutination test and HI test, so the HI titer was same as when used 0.5% red blood cells. Besides, based on our experience, we could get HI test results more quickly and clearly when use 0.75% red blood cells than 0.5%.

The HI titers had been added in figure 2A.

Point 7: Line 86: is there a mistake with the word MAGA X? perhaps MEGA X?

Response 7: The “MAGA X” had been changed to “MEGA X”

Point 8: Line 95: “were performed” instead of “was performed”?

Response 8: The “was performed” had been changed to “were performed”

3 - Results:

Point 9: Figure 1A: the legend of the figure A is incomplete (unit…)

Response 9: The figure legend had been completed.

Point 10: Line 124: can the authors explain the threshold of HI titer of 80 for considering positive serum? As HI titers are available upon request, can the authors give more information, i.e. the range of HI titers, the mean…

Response 10: According to WHO guideline for vaccine evaluation, the HI antibody titers ≥ 40 were considered as positive. But based on our experience, considering HI antibody titers ≥ 80 as positive would avoid the cross antigen reaction further. The HI titers information had been added in Figure 2A.

Point 11: Line 134: a comma missing

Response 11: The comma had been added.

Point 12: L135: plural sentence

Response 12: The sentence had been changed into plural form.

Point 13: L140-143: to be revised, not really agree. It’s hard to make a relationship between seroprevalence and positive detection without understanding the sampling of the active surveillance. Moreover, such explanation should be in the discussion part.

Response 13: The sampling details had been added in supplementary Table S2 and S3. More explanation had been added in the discussion part.

Point 14: Figure 3 and 4: the resolution must be improved because we cannot read the name of the viruses when enlarging the tree. The legend of the figure 4 must be revised. A sentence is written twice.

Response 14: The resolution of Figure 3 and 4 had been improved. The legend of Figure 4 had been revised.

Point 15: L158-165: please, revise the English language (singular/plural…)

Response 15: These sentences had been revised.

Point 16: L176-177: “due to its M gene”, without ‘s’. Why the authors did not appropriate the genotypes nomenclature by Sun [12]? G1 and G2 are not new in China, and correspond to G4 and G5 respectively. G1 is also the same genotype called G14 by Xu [13], isn’t it?

Response 16: The sentence had been revised.

According to Sun’s study, our G1 and G2 were corresponding to G4 and G5 respectively. And they declared that “G5 virus was detected continuously from 2013 to 2017, but it has declined since 2015 and was not found in 2018”.

But according to Xu’s study, our G2 was corresponding to G15. However, our G1 genotype was not found in Xu’s study. G1 was for the first time found in Liaoning.

Because of these differences, we decided to name the genotype as G1 and G2 in Liaoning Province 2020 SIV surveillance study. 

Point 17: L182: add ‘in’ in the sentence “… had been detected in the EA H1N1 SIVs.”

Response 17: The sentence had been revised.

Point 18: L179-180: specify that HP/LP is specific to the avian viruses adding, “which was characteristic of low pathogenic avian influenza virus”.

Response 18: The “avian” had been added.

Point 19: L184: add the reference to table 2 at the end of the sentence.

Response 19: “as shown in Table 2” had been added.

Point 20: Table 2: please modify Ha-E158G by HA-E158G

Response 20: Table 2 had been modified.

Point 21:  L211: move the comma after genes and not after NA

Response 21: The sentence had been revised.

Point 22: L215: remove “and positive” in this sentence?

Response 22: The sentence had been revised.

Discussion

Point 23: L241: modulate the idea with the conditional tense

Response 23: The sentences had been revised.

Point 24: L274-275: are there data of dN/dS obtained from other influenza viruses (other subtypes/lineages) to compare the values and discuss this level of selection pressure?

Response 24: Larger dN/dS value indicated the viral genome under higher selective pressure. For example, in “Evolutionary Dynamics and Global Diversity of Influenza A Virus. J Virol 2015, 89, 10993-11001”, the author compared the selective pressure of HA genes of canine H3N8 and avian H3N8 from 2000 to 2011. The average selection pressure was 0.324 for canine H3N8 and 0.089 for avian H3N8. So they declared that the canine virus had a faster evolution rate due to the larger selection pressure.

In another article “Genetic Evolution Characteristics of Genotype G57 Virus, A Dominant Genotype of H9N2 Avian Influenza Virus. Front Microbiol. 2021 Mar 3; 12: 633835”, Wang et al. also compared the positive selection pressures on HA and NA genes of H9N2 virus from 2007 to 2019. H9N2 viruses was divided into two time periods: 2007-2012 and 2013-2019. It was found that the positive selection pressures on HA and NA genes from 2013 to 2019 were stronger compared to those from 2007 to 2012.

In our study, we divided Liaoning SIVs into three time periods: 2006-2013 (P1), 2014-2016(P2) and 2020 (P3) and we found that the positive selection pressures on HA and NA genes during P1 and P3 time period were higher than P2 time period, indicating more frequent amino acid mutations occurrence in HA and NA genes of SIVs in P1 and P3.

Conclusion

Point 25: L282: twice “under”

Response 25: The sentence had been revised.

Reviewer 2 Report

The authors have conducted surveillance of Eurasian avian-like H1N1 swine Influenza viruses in Liaoning province of China. Prevalence, genetic and evolutionary properties have been researched and analyzed. Overall, this paper is well designed but need to be presented in a more clear and concise way. After careful consideration, I suggest reconsidering after major revision.

  1. English needs to be revised very carefully. For example, 33195 nasal swabs were collected from how many pigs? The authors might want to state as nasal swabs have been collected from 33195 pigs. Also check the abbreviations, numbers, auxiliary verbs, capitalizations, font, and contractions to avoid ambiguity.
  2. Figure 1A: why Dandong, Benxi, and Yingkou are not presented in the map? If there are no pig farms there, please state it in the content. Or the authors need to explain why they did not collect samples from these areas.
  3. PCR primers and conditions should be described in the materials and methods.
  4. Any Influenza vaccination data from these pig farms? If so, please include it in the manuscript.

Author Response

Response to Reviewer 2 Comments

Point 1: English needs to be revised very carefully. For example, 33195 nasal swabs were collected from how many pigs? The authors might want to state as nasal swabs have been collected from 33195 pigs. Also check the abbreviations, numbers, auxiliary verbs, capitalizations, font, and contractions to avoid ambiguity.

Response 1: English of the article had been revised.

Point 2: Figure 1A: why Dandong, Benxi, and Yingkou are not presented in the map? If there are no pig farms there, please state it in the content. Or the authors need to explain why they did not collect samples from these areas.

Response 2: According to National Bureau of Statistics of China data, the pig population of the 3 cities, Dandong, Benxi and Yingkou, took up only 6% of Liaoning pig population in total. The 3 cities were not the main pig breeding areas. Besides, we had no cooperative pig farms in these 3 cities for the sampling.

Point 3: PCR primers and conditions should be described in the materials and methods.

Response 3: The PCR condition had been added in the materials and methods part and the primers information had been added in the supplementary information part.

Point 4: Any Influenza vaccination data from these pig farms? If so, please include it in the manuscript.

Response 4: The influenza vaccine had not been applied in these pig farms.

Round 2

Reviewer 2 Report

Most of my questions have been answered. Thank you!

Author Response

Thank you for your advices.